# Delineating the transcriptional landscape and clonal diversity of virus-specific CD4+ T cells during chronic viral infection

Ryan Zander[1]*[†], Achia Khatun[1,2][†], Moujtaba Y Kasmani[1,2], Yao Chen[1,2], Weiguo Cui[1,2]*

[1]Blood Research Institute, Versiti Wisconsin, Milwaukee, United States; [2]Department of Microbiology and Immunology, Medical College of Wisconsin, Milwaukee, United States

**\*For correspondence:**
Ryan-Zander@uiowa.edu (RZ);
weiguo.cui@bcw.edu (WC)

[†]These authors contributed equally to this work

**Competing interest:** The authors declare that no competing interests exist.

**Abstract** Although recent evidence indicates that CD4+ T cells responding to chronic viral infection are functionally heterogenous, our understanding of the developmental relationships between these subsets, and a determination of how their transcriptional landscape compares to their acute infection counterparts remains unclear. Additionally, whether cell-intrinsic factors such as TCR usage influence CD4+ T cell fate commitment during persistent infection has not previously been studied. Herein, we perform single-cell RNA sequencing (scRNA-seq) combined with single-cell T cell receptor sequencing (scTCR-seq) on virus-specific CD4+ T cells isolated from mice infected with chronic lymphocytic choriomeningitis virus (LCMV) infection. We identify several transcriptionally distinct states among the Th1, Tfh, and memory-like T cell subsets that form at the peak of infection, including the presence of a previously unrecognized Slamf7+ subset with cytolytic features. We further show that the relative distribution of these populations differs substantially between acute and persistent LCMV infection. Moreover, while the progeny of most T cell clones displays membership within each of these transcriptionally unique populations, overall supporting a one cell-multiple fate model, a small fraction of clones display a biased cell fate decision, suggesting that TCR usage may impact CD4+ T cell development during chronic infection. Importantly, comparative analyses further reveal both subset-specific and core gene expression programs that are differentially regulated between CD4+ T cells responding to acute and chronic LCMV infection. Together, these data may serve as a useful framework and allow for a detailed interrogation into the clonal distribution and transcriptional circuits underlying CD4+ T cell differentiation during chronic viral infection.

## Editor's evaluation

The present study by Zander et al., aims at improving our understanding of CD4+ T cell heterogeneity in response to chronic viral infections. The authors utilize the murine LCMV c13 infection model and perform single-cell RNA seq analysis on day 10 post-infection to identify multiple, previously unappreciated, T-cell subsets and then go on and verify these analyses using multi-color flow cytometry before comparing the transcriptome of CD4 T cells from chronic infection to a previously generated data set of CD4 T cells obtained from acutely-resolved LCMV infection. This is a valuable transcriptomic resource of murine CD4 T cell subsets in chronic viral infection. The study will be of broad interest to a wide range of researchers focused on studying CD4 T cell biology.

## Introduction

CD4[+] T cells play an essential role in coordinating innate and adaptive immunity against multiple pathogens. Upon antigen encounter and exposure to inflammatory cytokines, CD4[+] T cells display an inherent capacity to adapt diverse functional fates and facilitate immunity through their secretion of soluble meditators and direct cell interactions with other immune cell populations. In response to either an acute or persistent viral infection, naïve antigen-specific CD4[+] T cells clonally expand and primarily differentiate into either T helper type 1 (Th1) or T follicular helper (Tfh) cells, which support cellular or humoral responses, respectively (*Crotty, 2011*; *Sheikh and Groom, 2021*).

Several recent studies have provided insight into how the fate commitment of CD4[+] T cells is established early after CD4[+] T cell priming. Notably, the strength of T cell receptor (TCR) and IL-2R signaling strongly influences this bifurcation decision, with increased TCR and IL-2R signaling generally favoring Th1 lineage commitment (*Choi et al., 2011*; *Pepper et al., 2011*; *Johnston et al., 2012*; *Snook et al., 2018*). Mechanistically, activation of the TCR and IL-2R signaling pathways results in the induction of Blimp-1, a transcription factor (TF) that is critical for Th1 differentiation and that also potently antagonizes Bcl-6, a transcriptional repressor that is required for Tfh differentiation (*Johnston et al., 2009*; *Choi et al., 2011*; *Pepper et al., 2011*). Additionally, early exposure of activated CD4[+] T cells to IL-12 and type I IFN signaling can promote Th1 cell differentiation by inducing expression of T-bet (*Hsieh et al., 1993*; *Manetti, 1993*; *Mullen et al., 2001*; *Ray et al., 2014*), which is the lineage-defining TF that is required for Th1 development (*Szabo et al., 2000*). Th1 cells can then travel to sites of infection where they produce pro-inflammatory cytokines such as IL-2, IFN-γ, and TNF-α, which contribute to the activation of CD8[+] T cells and macrophages (*Sheikh and Groom, 2021*). Thus, Th1 cells play a critical role in orchestrating immunity against intracellular pathogens. Conversely, dendritic cell presentation of antigen and secretion of IL-6 facilitates early Tfh differentiation by inducing Bcl-6 expression and the subsequent upregulation of CXCR5 (*Crotty, 2011*; *Eto et al., 2011*; *Choi et al., 2013*). Pre-Tfh cells then migrate from the T cell zone to the T cell-B cell border where they can interact with antigen-presenting B cells via ICOS-ICOSL and CD40-CD40L pathways, which are required for fully instilling the Tfh program (*Han et al., 1995*; *Choi et al., 2011*; *Crotty, 2011*). Mature Tfh cells then provide essential help signals to pathogen-specific B cells to facilitate germinal center (GC) reactions, B cell secretion of high-affinity antibodies, and the generation of memory B cells and long-lived plasma cells (*Crotty, 2011*).

After the initial T cell expansion phase, the pool of virus-specific CD4[+] T cells begins to decline in both self-resolving and chronic infections, although the nature of the infection plays a major role in shaping the overall magnitude, subset distribution, and functional capacity of responding CD4[+] T cells. In acute settings, following pathogen clearance, distinct populations of memory CD4[+] T cells develop that are well equipped to confer long-lasting protection against re-infection. These memory cells include Th1- and Tfh-like subsets (which may represent T effector memory cells; *Marshall et al., 2011*; *Pepper et al., 2011*; *Hale et al., 2013*), tissue resident memory cells that provide rapid protection at frontline sites of infection (*Iijima and Iwasaki, 2014*; *Schreiner and King, 2018*), and a less differentiated T central memory (Tcm) cell population that displays an increased capacity to proliferate and produce IL-2 upon secondary challenge (*Pepper and Jenkins, 2011*). While several studies have provided evidence that differentiated effector Th1 and Tfh cells can give rise to long-lived memory cells, previous work further indicates that a memory precursor population develops among early responding CD4[+] T cells and that this precursor population also significantly contributes to the establishment of durable and effective CD4[+] T cell memory following resolution of the infection (*Marshall et al., 2011*; *Pepper et al., 2011*; *Ciucci et al., 2019*). Notably, similar to that of Tcm cells, this precursor subset displays high expression of CCR7, TCF-1, and Bcl2, and as such was referred to as Tcm precursor (Tcmp) cells (*Ciucci et al., 2019*). Importantly, these Tcmp cells display an augmented capacity to survive the contraction phase of acute infection and generate diverse effector cell populations during a secondary response (*Marshall et al., 2011*; *Pepper et al., 2011*). The development and functional fitness of Tcmp cells have recently been identified to be dependent on the TF Thpok (*Ciucci et al., 2019*), although previous work also indicates that Bcl-6, in addition to its well-established role in facilitating Tfh differentiation, is also important for the generation of Tcmp cells (*Pepper et al., 2011*). Thus, CD4[+] T cell differentiation during acute viral infection is a well-synchronized process that results in appropriately balanced Th1, Tfh, and Tcmp cell formation at the peak of infection, and these subsets can then give rise to highly functional memory populations.

In contrast to acute settings, CD4+ T cells responding to persistent infection or cancer often acquire dysfunctional features, including a diminished capacity to produce pro-inflammatory cytokines that coincides with the upregulation of co-inhibitory receptors such as PD-1, Tim-3, and CTLA-4 (*Fuller et al., 2004*; *Brooks et al., 2005*; *Crawford et al., 2014*). Additionally, several studies have identified that Th1 formation and function is drastically reduced during chronic viral infection, and this decline in Th1 formation is accompanied by a relative increase in Tfh cell development (*Fahey et al., 2011*; *Yamada et al., 2016*). This shift in CD4+ T cell differentiation is thought to be important to limit Th1-mediated immunopathology (*Penaloza-MacMaster et al., 2015*), while at the same time promote Tfh-mediated antibody responses that are required for viral clearance from the periphery (*Fahey et al., 2011*; *Harker et al., 2011*; *Cook et al., 2015*; *Greczmiel et al., 2017*). Notably, recent work in the field has further demonstrated that in addition to the anticipated Th1 and Tfh populations that form during chronic viral infection, a previously unrecognized memory-like CD4+ T cell subset also develops, and this subset bears a transcriptional program similar to that of Tcmp cells from acute infection, as evident by its high expression of *Tcf7*, *Ccr7*, *Il7r*, and *Bcl2* (*Snell et al., 2021*; *Andreatta et al., 2022*; *Xia et al., 2022*; *Zander et al., 2022*). Moreover, memory-like CD4+ T cells were found to display an augmented capacity to accumulate during persistent infection, and they also exhibited the propensity to give rise to either Th1 or Tfh cells, indicating that this subset displays some developmental plasticity (*Xia et al., 2022*; *Zander et al., 2022*). Thus, CD4+ T cells responding to chronic viral infection are also capable of exhibiting memory-like properties, even in the presence of ongoing viral replication. However, despite these recent advances, our understanding of how CD4+ T cells adapt under settings of persistent antigen exposure and inflammation remains incompletely understood. Moreover, although recent reports have demonstrated that several transcriptionally distinct CD4+ T cell subsets develop during acute viral infection (*Ciucci et al., 2019*; *Khatun et al., 2021*), whether analogous or additional heterogeneous populations of CD4+ T cells emerge during chronic viral infection, and how these subsets compare to those from acute infection have remained significant knowledge gaps in the field. Additionally, questions remain about how chronic viral infection perturbs the functional and transcriptional landscape of CD4+ T cell populations, and whether these alterations in the CD4+ T cell compartment are due to global differences in gene expression programs or stem from differences in the distribution pattern of T helper cell subsets that develop under these contexts.

While previous studies have demonstrated that most antigen-specific CD4+ T cell clones can give rise to multiple lineages following acute bacterial or viral infection (*Tubo et al., 2013*), recent evidence indicates that some clones may also display a biased cell fate decision (*Khatun et al., 2021*). Importantly, cell-intrinsic factors, such as TCR signal strength, TCR affinity for cognate peptide, and TCR chain usage have all recently been implicated in contributing to shaping CD4+ T cell fate decision during viral infection (*Tubo et al., 2013*; *Tubo et al., 2016*; *Cho et al., 2017*; *Künzli et al., 2020*; *Khatun et al., 2021*). Interestingly, TCR signal strength was found to exert opposite effects on the balance of Th1 to Tfh cells depending on whether the infection was acutely resolving or persistent (*Künzli et al., 2020*). However, we currently do not fully understand the extent by which TCR repertoire impacts cell fate decisions during chronic viral infection. Additionally, whether naïve T cells with particular TCR chains display the propensity to give rise to multiple independent lineages during chronic viral infection, or whether some clones preferentially acquire a particular differentiation program that may skew the T helper cell response in the setting of chronic infection, remains unclear.

In this study, we employed single-cell RNA sequencing (scRNA-seq) coupled with single-cell TCR sequencing (scTCR-seq) to investigate the transcriptional landscape of CD4+ T cells responding to chronic lymphocytic choriomeningitis virus (LCMV) infection. Notably, we uncovered previously unappreciated transcriptional heterogeneity among the virus-specific CD4+ T cell pool, including the identification of several transcriptionally distinct states within the Th1, Tfh, and T memory-like cell populations. Moreover, we found that, similar to findings from acute viral infection, the majority of CD4 T cell clones responding to chronic viral infection displayed remarkable developmental plasticity as evidenced by their membership across multiple distinct subsets, although a small fraction of clonotypes did appear to exhibit a preferred lineage choice. Lastly, using an integrated analysis to compare the transcriptional program and subset distribution of CD4+ T cells responding to either acute or chronic LCMV infection, we identified that persistent infection is not only associated with altered CD4+ T cell subset formation and gene expression profiles, but also changes in core gene expression modules that span multiple distinct T helper cell populations.

## Results

## CD4⁺ T cells responding to chronic LCMV infection are transcriptionally diverse and are dominated by a few large clones

To better understand the transcriptional heterogeneity and clonal diversity among CD4 T cells responding to chronic viral infection, we performed paired scRNA-seq and scTCR-seq on splenic $GP_{66-77}$ ($GP_{66}$)-specific CD4 T cells that were isolated from two individual mice on day 10 post-LCMV Clone 13 (Cl13) infection, a timepoint in which the endogenous $GP_{66}$-specific T cell response is approximately at its peak (*Crawford et al., 2014*). We defined a clone as a group of T cells sharing the same nucleotide sequence from the CDR3 regions of both the α and β chains of the TCR. In order to safeguard against potential errors produced during fluorescence-activated cell sorting or sequencing misreads, we restricted our analysis to clones with at least two cells sharing the same TCR α and β chain CDR3 sequences. This resulted in a total of 168 clones for mouse 1 (M1) and 159 clones for mouse 2 (M2) (*Figure 1A*), consisting of 5185 total cells between our two samples (2836 cells for M1 and 2349 cells for M2). These findings are in line with the expected range of TCR clonotypes reported in mice (*Kotturi et al., 2008*; *Jenkins and Moon, 2012*; *Khatun et al., 2021*) and suggest that our scRNA-seq approach likely recovered the majority of the constituents from the $GP_{66}$-specific clonal repertoire. Interestingly, and similar to our analysis on the T cell repertoire for $GP_{66}$-specific CD4⁺ T cells during acute LCMV Armstrong infection, we did not observe any clonal overlap between M1 and M2 when α and β chain CDR3 sequences were paired on a per cell basis, although a few instances of TCR CDR3 overlap were observed when considering either α and β chain CDR3 sequences individually (*Figure 1B*). These results highlight a tremendous amount of diversity within the TCR repertoire, even among genetically identical hosts. We next assessed patterns of clonal distribution and dominance. Notably, the top five clones for each mouse (which ranged from comprising 3.7 to 9.8% of the $GP_{66}$-specific pool) accounted for ~40–45% of all cells in the dataset (*Figure 1A and C*), demonstrating that the CD4⁺ T cell response during LCMV Cl13 infection is dominated by a few large clones.

Next, using uniform manifold approximation and projection (UMAP) to visualize single-cell gene expression data, we identified the presence of eight transcriptionally distinct clusters stemming from our dataset containing ≥2 cells per clone of $GP_{66}$-reactive CD4⁺ T cells (*Figure 1D*, *Figure 1—figure supplement 1A*). Of these eight clusters, clusters 2 and 5 both displayed increased expression of several Th1 cell-associated genes, such as *Cxcr6*, *Tbx21* (encodes T-bet), *GzmB*, and *Id2* (*Shaw et al., 2016*; *Choi et al., 2020*; *Figure 1E–F*), suggesting that these two populations are of the Th1 lineage. Interestingly, cluster 2 displayed increased expression of *Ly6c2*, *Klf2*, and *S1pr1* and also displayed uniquely high expression of *Cx3cr1*, whereas cluster 5 displayed increased expression of *Lag3*, *Slamf1*, *Pdcd1*, and *Tnf* (*Figure 1E*, *Figure 1—figure supplement 1B*). These findings may suggest the presence of functional heterogeneity within the Th1 pool, and we henceforth refer to these populations as the Ly6c-hi Th1 and Lag3-hi Th1 clusters, respectively. Using Seurat's module score feature to more accurately quantify the gene expression patterns of these clusters in comparison to published gene sets further supported that clusters 2 and 5 are composed of Th1 cells (*Figure 1I*). By contrast, clusters 4 and 7 displayed the lowest Th1 module scores and also displayed increased expression of Tfh-associated genes such as *Cxcr5* and *Bcl6* (*Figure 1E,G,I*). Cluster 4 in particular displayed high expression of genes important for GC Tfh differentiation and function, including *Ascl2*, *Tox2*, and *Il4* (*Crotty, 2014*; *Liu et al., 2014*; *Xu et al., 2019*; *Figure 1G*). Consistent with these observations, cluster 4 displayed the highest Tfh module score (*Figure 1I*), followed by cluster 7, the latter of which also displayed a transcriptional profile similar to the T-bet–expressing Tfh subset that develops during acute LCMV Armstrong infection (*Khatun et al., 2021*; *Figure 1—figure supplement 1C*). Thus, we referred to clusters 4 and 7 as GC Tfh2 and Tfh1 cell subsets, respectively.

Clusters 0 and 1 both exhibited high expression of *Slamf6* (encodes Ly108), as well as multiple memory-associated genes, including *Tcf7*, *Il7r*, *Ccr7*, and *Klf2* (*Ciucci et al., 2019*; *Figure 1E and H*). Consistent with this observation, clusters 0 and 1 also displayed increased memory T cell module scores (*Figure 1I*) and relatively lower Th1 and Tfh module scores, although cluster 1 did display the third highest Tfh module score behind the GC Tfh2 and Tfh1 clusters. Together, these data suggest that cells from clusters 0 to 1 likely fall within the *Slamf6*⁺ T memory-like cell subset that we and others have recently demonstrated develops during chronic LCMV infection, of which largely remains in a quiescent-like state, yet does display some developmental plasticity (*Xia et al., 2022*; *Zander et al., 2022*). Interestingly, compared to cluster 0, cluster 1 did display increased expression of *Tnfrsf4*, *Batf*,

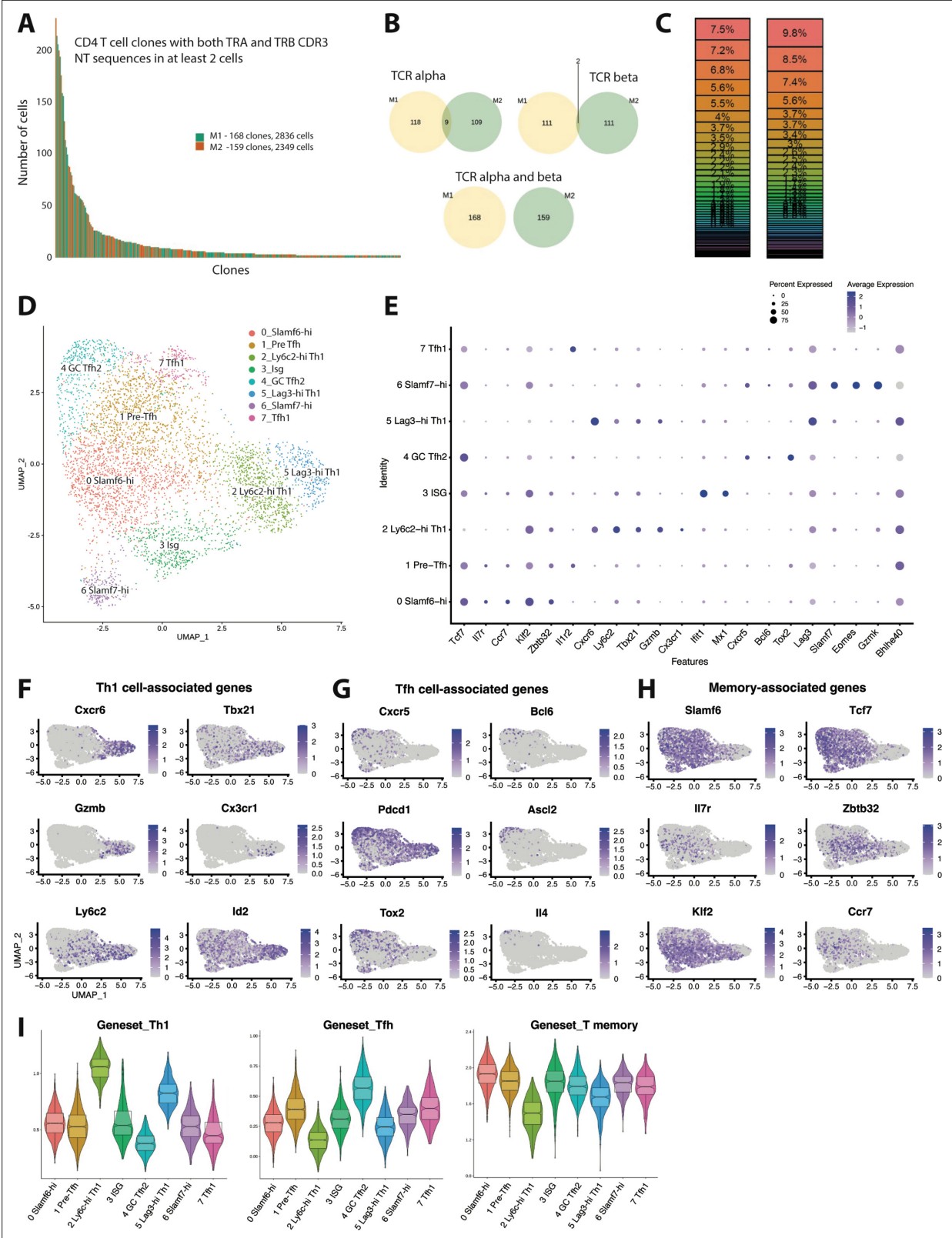

**Figure 1.** Clonal distribution and transcriptional landscape of GP66-specific CD4[+] T cells during chronic lymphocytic choriomeningitis virus (LCMV) infection. (**A–I**) GP66-specific splenic CD4 T cells were isolated and sort-purified from two individual LCMV Clone 13 (Cl13)-infected mice on day 10 post-infection, and the 10× Genomics pipeline was used to generate a combined single cell RNA-sequencing (scRNA-seq) + T cell receptor (TCR) seq library. A downstream computational analysis was then performed in R. (**A**) Summary data showing the size of and clonal distribution of GP66-specific CD4[+]

*Figure 1 continued on next page*

*Figure 1 continued*

T cells (with at least 2 cells per clone) recovered from the two separate mice. (**B**) Pie charts showing the shared number of public clones when assessing the sequence of either TCR alpha chains, TCR beta chains, or both TCR alpha and beta chains. (**C**) Summary data showing the relative frequency of each T cell clone recovered from the two mice. (**D**) Uniform manifold approximation and projection (UMAP) plot showing the presence of eight transcriptionally distinct clusters of GP66-specific CD4 T cells. (**E**) Dot plot showing the relative expression of differentially expressed genes between the eight clusters. (**F–H**) UMAP plots depicting the expression of either T helper type 1 (Th1) cell-associated genes (**F**), T follicular helper (Tfh) cell-associated genes (**G**), or memory-associated genes (**H**). (**I**) Module score analyses were performed to compare the relative Th1, Tfh, and memory T cell signatures among the distinct clusters of GP66-specific CD4$^+$ T cells.

The online version of this article includes the following figure supplement(s) for figure 1:

**Figure supplement 1.** CD4$^+$ T cells responding to chronic viral infection are composed of several transcriptionally and functionally distinct populations.

*Il1r2*, and *Cd83* (*Figure 1E*, *Figure 1—figure supplement 1A*), which were also highly expressed in the Tfh1 population, possibly indicating that cells from cluster 1 are in a more activated state. Given that cluster 1 had an increased Tfh module score relative to cluster 0, coupled with recent reports that some memory-like CD4$^+$ T cells can give rise to Tfh cells (*Xia et al., 2022*; *Zander et al., 2022*), we elected to refer to clusters 0 and 1 as Slamf6-hi and pre-Tfh cells respectively.

Lastly, our scRNA-seq analysis identified the presence of two additional transcriptionally distinct clusters, clusters 3 and 7, with cluster 3 cells displaying uniformly high expression of multiple type I IFN stimulated genes (ISGs), including *Ifit1*, *Ifit3*, *Isg15*, *Isg20*, *Mx1*, and *Cxcl10* (*Figure 1E*, *Figure 1—figure supplement 1A*). On the other hand, cluster 7 cells displayed uniquely high expression of *Slamf7*, *Eomes*, and *Gzmk* (*Figure 1E*, *Figure 1—figure supplement 1A*), and as such, cells within this cluster bore a resemblance to a cytolytic CD4$^+$ T cell subset that has been identified in tumors as well as autoimmune disease (*Della-Torre et al., 2018*; *Cachot et al., 2021*). Notably, and consistent with their diverse transcriptional programs, the eight clusters we identified also substantially differed in their relative expression of several cytokines and chemokines (*Figure 1—figure supplement 1D*). Although *Ifng* was expressed by all clusters, Ly6c-hi Th1 and Lag3-hi Th1 clusters displayed the highest expression of *Ifng*, and these respective clusters also had increased expression of the Th1-associated chemokine ligands (Ccl) *Ccl3*, *Ccl4*, and *Ccl5* (*Figure 1—figure supplement 1D*). Conversely, and consistent with our previous report (*Zander et al., 2022*), *Il21* expression was increased in the pre-Tfh, Tfh1, and GC Tfh2 populations (*Figure 1—figure supplement 1D*), whereas *Cxcl10*, a chemokine known to be induced by type I IFN signaling (*Groom and Luster, 2011*), was enriched in the ISG cluster. Intriguingly, the Slamf7-hi subset displayed increased expression of the immunoregulatory chemokine *Tgfb1*, although all eight clusters expressed *Tgfb1* to some extent (*Figure 1—figure supplement 1D*). Taken together, these data uncover broad transcriptional and functional heterogeneity among CD4$^+$ T cells responding to chronic LCMV infection.

## Validation of CD4$^+$ T cell heterogeneity during chronic LCMV infection

Using several differentially expressed surface markers identified in our scRNA-seq analysis, we next performed spectral flow cytometry to further investigate the phenotypic heterogeneity within the CD4$^+$ T cell compartment during LCMV Cl13 infection. Using a panel of 18 parameters, t-distributed stochastic neighbor embedding (t-SNE) visualization was performed on PD-1$^{hi}$ CD4$^+$ T cells (which include both GP$_{66}$-specific and other LCMV epitope-reactive CD4$^+$ T cells) that were obtained from LCMV Cl13-infected Mx1-green fluorescent protein (GFP) reporter mice. Notably, several surface molecules were differentially expressed amongst the pool of LCMV-specific CD4$^+$ T cells, which allowed for the formulation of several distinct clusters (*Figure 2A*). Consistent with our scRNA-seq analyses, a subset of cells displayed low expression of Ly108 and high expression of CXCR6, indicating that these cells are likely of the Th1 lineage (*Figure 2A*). Additionally, one small population (which only comprised ~2–6% of the CD4 T cell pool) displayed uniformly and exclusively high expression of Slamf7 (*Figure 2A*), indicating that these cells likely correspond with the *Slamf7*-hi cells from cluster 6 in our scRNA-seq analysis. Another subset displayed increased expression amounts of CXCR5 and PD-1 relative to the other cells, suggesting that this population is likely composed of cells found within the GC Tfh2 subset. Interestingly, some cells that clustered together exhibited intermediate expression amounts of CXCR5 and high expression of IL-1R2 (*Figure 2A*), a marker that was increased in both the pre-Tfh and Tfh1 subsets, possibly indicating that IL-1R2 expression can be used to distinguish these populations of CD4$^+$ T cells. A large proportion (~36%) of cells expressed high amounts of Ly108

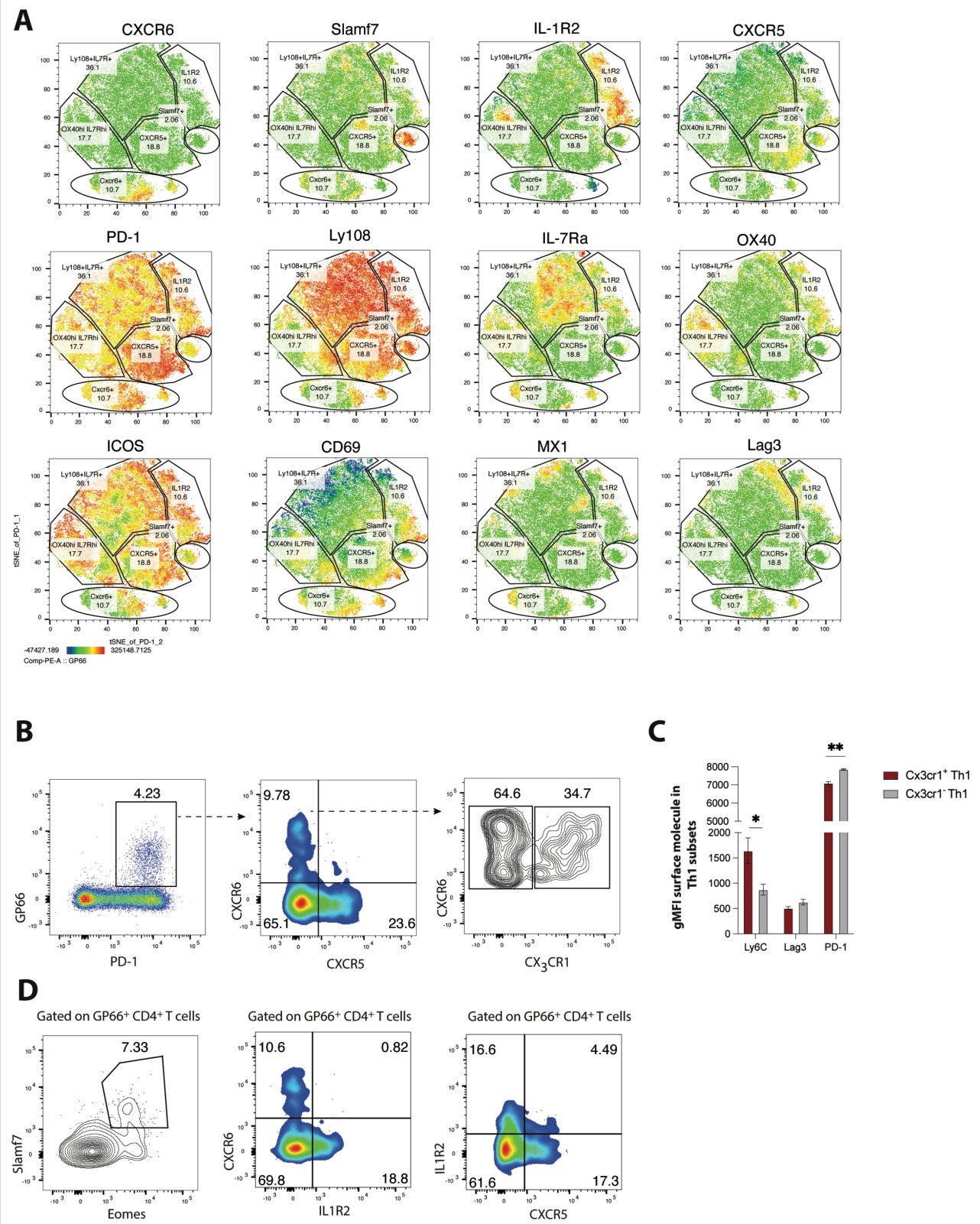

**Figure 2.** Validation of heterogeneity in CD4 T cells responding to chronic lymphocytic choriomeningitis virus (LCMV) infection using spectral flow cytometry. (**A**) MX1-GFP reporter mice were infected with LCMV Clone 13 (Cl13), and spectral flow cytometry was used to assess the phenotype of splenic CD4 T cells on day 25 p.i. Using a panel of 18 parameters, t-distributed stochastic neighbor embedding (t-SNE) visualization was performed on total PD-1hi CD4 T cells (which include both GP66-specific and other LCMV epitope-reactive CD4 T cells). Representative t-SNE flow plots in A are

*Figure 2 continued on next page*

*Figure 2 continued*

displayed for one mouse, with data being representative of three individual mice. (**B**) Representative flow plots showing potential gating strategy to identify CX3CR1 ± CXCR6+T helper type 1 (Th1) cell subsets. (**C**) Summary data showing the relative expression of indicated surface molecules in CX3CR1+ and CX3CR1− Th1 cell subsets. (**D**) Representative flow plots showing potential gating strategies to identify Slamf7+Eomes + (far left), and IL-1R2+CD4 T cell subsets (which may be composed of pre-T follicular helper [Tfh] and Tfh1 subsets). Data (mean ± SEM) in **C** are from three individual mice, and were analyzed with unpaired student's t-tests. Data are representative of at least two independent experiments. * p < 0.05, ** p < 0.01.

and IL-7Ra, suggesting that this population of cells likely corresponds with the *Slamf6*-hi cells from cluster 0. Of note, cells did not appear to cluster together based off of Mx1-GFP expression; rather, Mx1, an IFN-stimulated gene, was detected via GFP expression across all t-SNE clusters (*Figure 2A*). Given that type I IFN-stimulated genes remain elevated for several weeks following chronic viral infection (*Teijaro et al., 2013*; *Wilson et al., 2013*), this finding suggests that all subsets of CD4+ T cells are likely to experience some degree of type I IFN signaling during persistent viral infection. In contrast to our findings at the Mx1-GFP protein level, we likely detected a discrete cluster of IFN-stimulated CD4+ T cells by scRNA-seq because IFN-stimulated genes transiently dominated the transcriptomes of certain cells, causing them to cluster together.

Recently, we demonstrated that differential expression of CXCR6 and CXCR5 can be used to distinguish three major CD4+ T cell subsets that differ in their phenotype, function, transcriptional program, and epigenetic landscape: CXCR6$^{hi}$ CXCR5$^{lo}$ CD4 T cells exhibit a robust Th1 cell program, CXCR6$^{lo}$ CXCR5$^{hi}$ CD4 T cells correspond with Tfh cells, and CXCR6$^{lo}$ CXCR5$^{lo}$ cells (which have high Ly108, IL-7R, and TCF-1 expression) display a memory-like signature and as such were defined as a T memory-like cell subset (*Zander et al., 2022*). Given the additional heterogeneity uncovered in this study, we next aimed to determine some potential gating strategies that can be used to further distinguish some of the major CD4+ T cell subsets identified in our scRNA-seq analyses. Notably, and consistent with our scRNA-seq findings, GP$_{66}$-specific CXCR6+ Th1 cells could be further stratified based on CX$_3$CR1 expression, with CX$_3$CR1$^{hi}$ CXCR6$^{hi}$ cells displaying increased expression of Ly6C and lower expression of PD-1 compared to CX$_3$CR1$^{lo}$ CXCR6$^{hi}$ cells (*Figure 2B–C*). Given that distinct Th1 subsets can emerge during different contexts and have diverse functional roles (*Krueger et al., 2021*), a more in-depth investigation into the functional heterogeneity of these Th1-like populations may have important implications for diseases where Th1 cells play a dominant role.

Using a different gating strategy, we detected a minor population of GP$_{66}$-specific CD4+ T cells that displayed high co-expression amounts of Eomes and Slamf7 (*Figure 2D*), with these markers being largely unique to the *Slamf7*$^{hi}$ subset identified in our scRNA-seq analysis. Thus, high expression of Slamf7 may be a reliable surface marker for this particular subset of CD4+ T cells. Similarly, a clear population of IL-1R2-expressing CD4+ T cells was detected amongst the GP$_{66}$-specific pool (*Figure 2D*), with the majority of these cells staining negative for CXCR6 and also exhibiting low CXCR5 expression, possibly suggesting that this expression module may potentially mark the pre-Tfh subset. Interestingly, a small fraction (~4–5%) of GP$_{66}$-specific CD4 T cells coordinately expressed both IL-1R2 and CXCR5, possibly marking cells from the Tfh1 cluster (*Figure 2D*), which our scRNA-seq analyses identified as expressing both of these molecules (*Figure 1E and G*). Collectively, our results highlight several potential gating strategies and markers that may be useful in demarcating the populations identified in our scRNA-seq analysis, which may allow for further characterization of these transcriptionally and phenotypically distinct subsets.

## Trajectory analysis at the clonal level demonstrate developmental plasticity among most GP$_{66}$-specific CD4+ T cell clones during chronic LCMV infection, although some clones do display a biased-cell fate decision

To gain insight into the potential developmental relationships between these cell subsets, we employed use of the Monocle package in R (*Qiu et al., 2017*). Monocle aligns cells based on gene expression levels to determine cell trajectories in psuedotime, a measure of how much progress an individual cell has made along a differentiation pathway (*Qiu et al., 2017*). Our Monocle analysis formed a tree structure containing three major branches, with cells from the Ly6c-hi Th1 and Lag3-hi Th1 clusters residing on the far-right branch, cells from the GC Tfh2, Tfh1, and Slamf7 clusters dwelling on the far-left branch (as well as on one minor branch stemming off of this major branch), and cells from the Slamf6-hi

subset localizing more on the central branch (*Figure 3A*). Notably, cells from both the Slamf6-hi and pre-Tfh clusters could readily be detected along all three branches (*Figure 3A–B*), possibly indicating that these subsets may retain a higher degree of developmental plasticity. Conversely, the position of Th1 cells and Tfh cells at the far ends of the branches may insinuate that these subsets have reached a more terminal stage of differentiation. Interestingly, whereas ~11.5% of Slamf6-hi cells were found to reside along the Tfh branch, ~42.6% of cells from the pre-Tfh cluster were found along this branch (*Figure 3A–B*), further supporting that cells from the pre-Tfh cluster may be en route toward developing along a Tfh differentiation pathway. Notably, findings from our Monocle trajectory analysis closely align with recent reports demonstrating that memory-like CD4$^+$ T cells are capable of giving rise to both Tfh and Th1 effector cells subsets following adoptive transfer (possibly through an intermediate state), whereas Th1 and Tfh cells do not readily interconvert between one another during chronic viral infection (*Xia et al., 2022*; *Zander et al., 2022*).

Previous work indicates that in response to acute viral or bacterial infection, the majority of pathogen-specific CD4$^+$ T cell clones can give rise to both Th1 and Tfh cells following clonal expansion (*Tubo et al., 2013*; *Becattini et al., 2015*; *Cho et al., 2017*). Accordingly, we recently reported that a sizeable fraction (~28%) of CD4$^+$ T cell clones display a preferred lineage choice toward either Th1 or Tfh cells during acute LCMV Armstrong infection (*Khatun et al., 2021*). Notably, TCR structure, and in particular the CDR3 motif of the TCR α chain, was determined to be a driving influencer of this biased cell fate decision (*Khatun et al., 2021*). However, given that TCR signaling strength may exert opposite effects on the balance of Th1 to Tfh cells depending on the nature of the infection (*Künzli et al., 2020*), whether CD4$^+$ T cells responding to chronic viral infection also display similar patterns of differentiation at the clonal level remains unclear. Merging our single-cell clonal and transcriptomic datasets together allowed us to track the lineage commitment of CD4$^+$ T cell clones from the GP$_{66}$-specific repertoire during chronic LCMV Cl13 infection. Notably, a high degree of clonal overlap was observed amongst all CD4$^+$ T cell subsets identified in our scRNA-seq analysis, with most subsets displaying >30% overlap with any given subset (*Figure 3C*). In line with this finding, upon examination of the top five clones (which contained 193–230 cells per clone), we identified that each clone displayed membership in every cluster from our scRNA-seq analysis (*Figure 3D*), suggesting that similar to acute infection, one naïve CD4$^+$ T cell displays the capacity to give rise to multiple T helper cell lineages during persistent infection. Intriguingly, when assessing the cellular distribution of all clones with at least 2 cells per clone (168 clones for M1 and 159 clones for M2), we identified that while the majority of clones (~75%) had some membership in at least two or more lineages (Th1, Slamf6-hi, or Tfh as defined by Monocle branch), a small fraction of clonotypes exhibited a preferred lineage choice, with ~15% of clones displaying select membership in only the Th1 lineage and <10% of clones displaying exclusive membership in either the Slamf6-hi or Tfh lineages (*Figure 3E*). To more formally test whether some CD4$^+$ T cell clones display a biased cell fate decision during chronic LCMV infection, we performed a more stringent analysis in which we restricted our dataset to clones that have at least four cells (176 clones between two mice), and Th1 or Tfh-biased clones were then identified by assessing the lineage membership of each clone (based on their Monocle state) using a threshold cut-off of 65% (*Khatun et al., 2021*). Notably, and similar to our analysis using at least 2 cells per clone, we identified that the majority of GP$_{66}$-specifc clones appear multi-fated (*Figure 3—figure supplement 1*, and *Supplementary file 1*). Importantly, however, a small proportion of clones did display a biased cell fate decision, with approximately 21% of the 176 clones displaying a preferential skewing toward the Th1 lineage (p=2.443521e-09) and 8.5% showing a preferential bias for a Tfh cell fate (p=3.903672e-05; Figure S2B-C). Thus, similar to our observations in acute LCMV infection (*Khatun et al., 2021*), while the majority of GP$_{66}$-specific CD4$^+$ T cell clones exhibit some developmental plasticity, a minor proportion of clones do appear to display a biased cell-fate decision.

## CD4$^+$ T cells responding to chronic LCMV infection display altered subset distribution patterns and core changes in their transcriptional program

Global (bulk) transcriptional analysis previously performed on total GP$_{66}$-specific CD4$^+$ T cells isolated from mice either acutely or persistently infected with LCMV identified distinct patterns of gene expression, with CD4$^+$ T cells responding to chronic infection displaying increased expression of genes encoding PD-1, Lag3, and CTLA-4 co-inhibitory receptors, as well as increased expression of

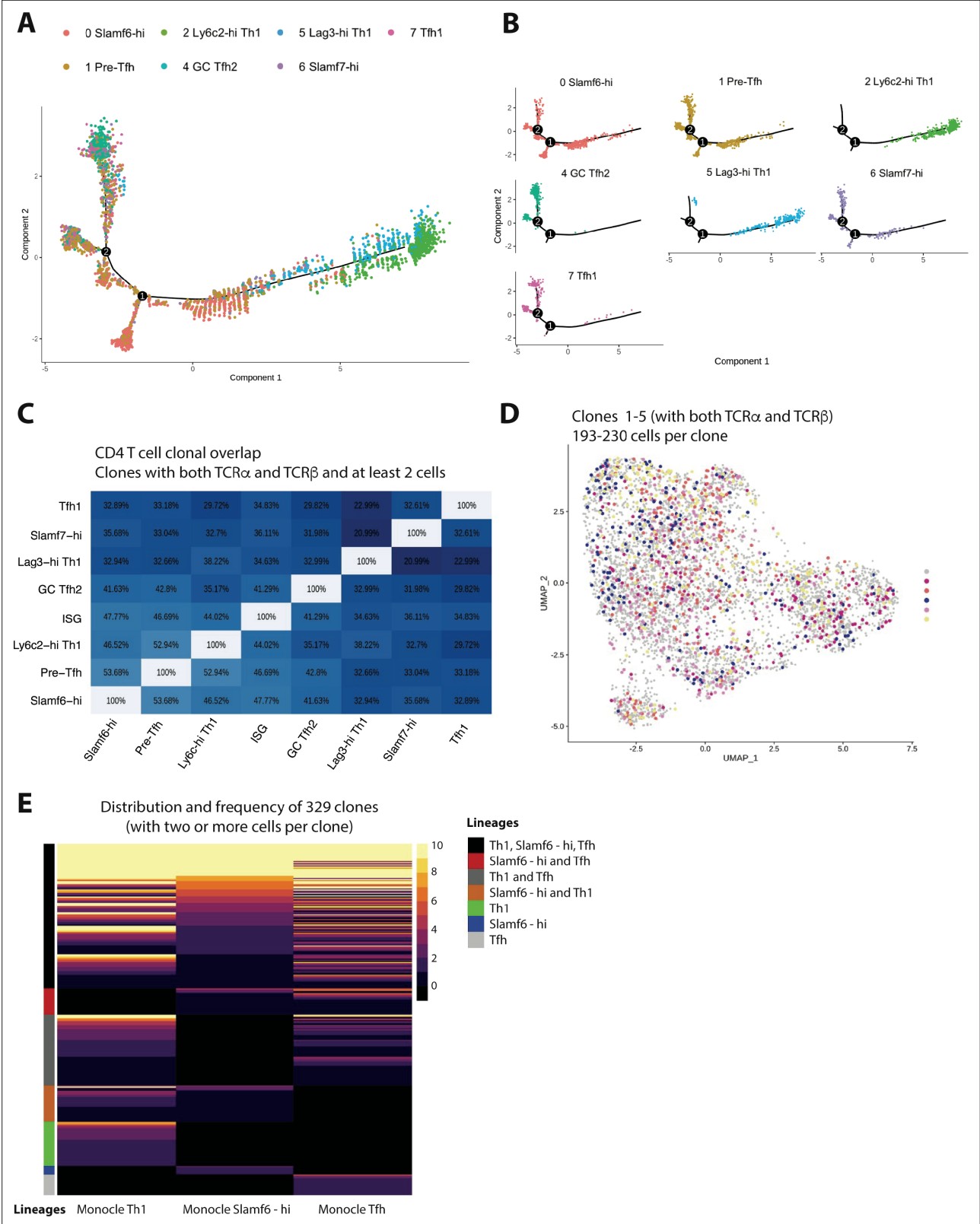

**Figure 3.** Trajectory analysis and clonal lineage tracing highlights that most GP66-specific CD4 T cells display developmental plasticity during chronic lymphocytic choriomeningitis virus (LCMV) infection, although a small fraction of clones exhibit a biased cell fate decision. (**A and B**) Monocle tree trajectory plots showing predicted cellular differentiation of GP66-specific CD4 T cell subsets based on trajectory analysis. (**C**) Summary chart displaying clonal overlap among GP66+ CD4 T cell subsets. (**D**) Uniform manifold approximation and projection (UMAP) plot as in **Figure 1D**, but colored by clone

*Figure 3 continued on next page*

*Figure 3 continued*

membership. (**E**) Heatmap showing subset distribution and frequency of 329 clones (n=168 clones from mouse 1 [M1] and n=159 clones from mouse 2 [M2]) among indicated CD4 T cell lineage (T helper type 1 [Th1], T follicular helper [Tfh], and Slamf6$^{hi}$ memory) as defined by Monocle state.

The online version of this article includes the following figure supplement(s) for figure 3:

**Figure supplement 1.** Trajectory analysis and clonal lineage tracing indicate that a small fraction of GP66-specific CD4 T cells display display a biased cell fate decision.

---

several TFs, including *Eomes*, *Prdm1* (encodes Blimp-1), *Zbtb32*, and *Klf4* (*Crawford et al., 2014*). However, as multiple transcriptionally distinct subsets have since been found to develop during acute and chronic viral infection, whether these observed alterations in the transcriptional program of CD4$^+$ T cells stem from differences in the relative distribution of CD4$^+$ T cell subsets that develop under different contexts or from a conserved gene expression program acquired by CD4$^+$ T cells responding to chronic viral infection remains unclear. To gain insight into this question, we performed a comparative scRNA-seq transcriptomic analysis by combining our previously published scRNA-seq data of GP$_{66}$$^+$ CD4$^+$ T cells isolated from LCMV Armstrong-infected mice on day 10 post infec (*Khatun et al., 2021*) with our current dataset of GP$_{66}$$^+$ CD4 T cells from day 10 post-LCMV Cl13 infection. To do this, we used the integration function in Seurat to perform a canonical correlation analysis (CCA) (*Butler et al., 2018*) and identify shared subpopulations across datasets. Notably, all of the clusters initially identified in our LCMV Cl13-restricted dataset (*Figure 1*) were also detected in this integrated dataset, although clear differences in the relative proportion of each cluster were observed when grouped by infection status (acute vs chronic; *Figure 4A*). In particular, the proportion of Ly6c2-hi Th1 and Lag3-hi Th1 cell subsets were dramatically reduced (by ~50%) during LCMV Cl13 as compared to LCMV Armstrong infection; this reduction in Th1 cell formation was accompanied by increased frequencies of Slamf6-hi, pre-Tfh, ISG, and Slamf7-hi clusters, whereas GC Tfh2 and Tfh1 clusters remained similar between infections (*Figure 4A–B*).

This observation is in line with previous work demonstrating a rapid loss of Th1 cell formation and function during chronic viral infection (*Fahey et al., 2011*; *Yamada et al., 2016*), as well as our recent finding that this decline in CXCR6$^+$ Th1 cell accumulation is accompanied by an increase in the formation of a Slamf6$^+$ memory-like CD4$^+$ T cell subset (*Zander et al., 2022*). In addition, our finding that the ISG cluster is overrepresented during LCMV Cl13 infection is consistent with previous reports showing that type I IFN-stimulated genes remain elevated during chronic viral infection compared to acute infection (*Teijaro et al., 2013*; *Wilson et al., 2013*). Interestingly, although Eomes was previously identified to be overexpressed in CD4$^+$ T cells during chronic infection (*Crawford et al., 2014*), in this study, we identified that this is likely due to a preferential expansion of the Slamf7$^+$ subset, of which displays uniquely high expression of Eomes (*Figure 4J*). Consistent with this observation, we found that the proportion of Slamf7$^+$ Eomes$^+$ virus-specific CD4$^+$ T cells was increased by ~10-fold during chronic LCMV Cl13 infection compared to acute LCMV infection (*Figure 4—figure supplement 1A*). Taken together, these data indicate that the nature of the infection (i.e. resolution vs chronicity) may impact the relative distribution of CD4$^+$ T cell subsets that develop with persistent infection resulting in a notable attenuation in Th1 responses and an accompanying increase in the formation of memory-like, pre-Tfh, ISG, and Slamf7$^+$ CD4$^+$ T cell populations.

To assess potential shifts in gene expression profiles that varied by infection and that spanned across all CD4$^+$ T cell subsets, we performed module score analyses to compare Th1, Tfh, memory T cell, TCR signaling, and T cell dysfunction gene signatures across all of the CD4$^+$ T cell clusters. Notably, module score analysis demonstrated that most CD4$^+$ T cell clusters from chronic LCMV infection exhibited decreased expression of the Th1-associated program, possibly suggesting that an attenuation in Th1-like activity is a broad feature of CD4$^+$ T cells responding to chronic viral infection and not only a result of a decrease in the Th1 lineage (*Figure 4C*, *Figure 4—figure supplement 1F*). Conversely, Tfh module scores were generally similar between acute and chronic LCMV CD4$^+$ T cell clusters, although the Ly6c2-hi Th1 and Lag3-hi Th1 cell subsets from chronic LCMV infection did display a relatively increased Tfh signature compared to their acute infection counterparts (*Figure 4D*, *Figure 4—figure supplement 1F*). Intriguingly, although the relative score varied by cluster, both memory T cell and Tcmp gene signatures, as well as progenitor CD8$^+$ T cell genes, were enriched across all CD4$^+$ T cell clusters stemming from chronic LCMV infection (*Figure 4E*, *Figure 4—figure supplement 1B-D*, 1 F), possibly suggesting that persistent infection promotes a core quiescence- or memory-associated

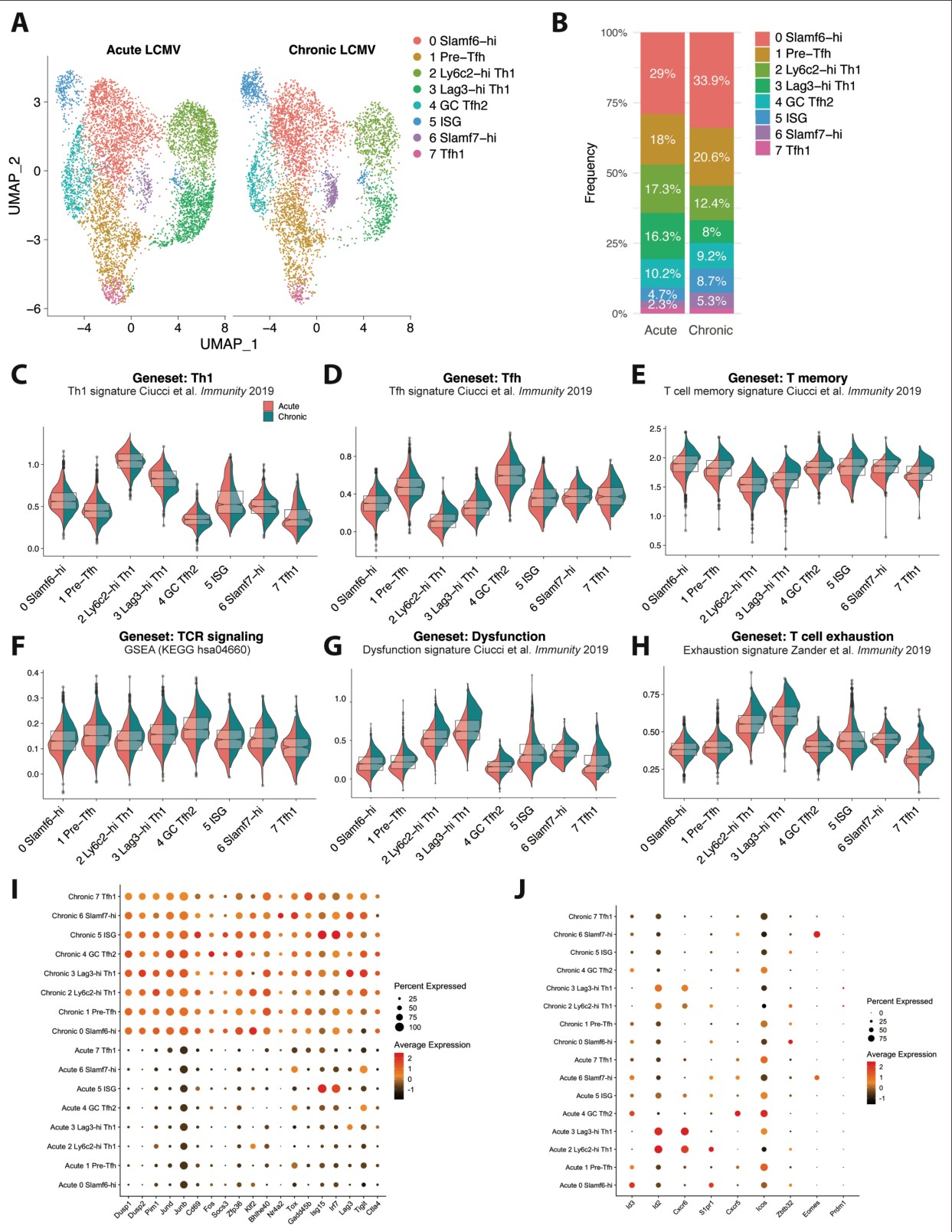

**Figure 4.** CD4 T cells responding to chronic viral infection display altered subset distribution patterns and core gene expression modules compared to their acute infection counterparts. (**A**) A canonical correlation analysis (CCA) was performed on single cell RNA-sequencing (scRNA-seq) libraries obtained from GP66-specific CD4+ T cells during chronic lymphocytic choriomeningitis virus (LCMV) Clone 13 (Cl13) infection (this study) and acute LCMV Armstrong infection (**Khatun et al., 2021**) that were integrated together using the integration function in Seurat (**Butler et al., 2018**).

*Figure 4 continued on next page*

*Figure 4 continued*

(**B**) Summary graph showing the relative subset distribution of GP66 + CD4 T cells subsets during acute and chronic LCMV infection on day 10 p.i. (**C–H**) Module score analyses showing the average expression levels of the T helper type 1 (Th1), T follicular helper (Tfh), memory T cell, T cell receptor (TCR) signaling, and T cell dysfunction programs among the CD4 T cell subsets identified in A. (**I–J**) Summary dot plots showing the average gene expression and the proportion of each cell subset expressing indicated gene. All genes listed in (**I**) were among the top 50 differentially expressed genes that were upregulated across multiple distinct subsets in response to chronic LCMV infection.

The online version of this article includes the following figure supplement(s) for figure 4:

**Figure supplement 1.** CD4 T cells responding to chronic lymphocytic choriomeningitis virus (LCMV) infection display altered subset distribution patterns and changes in gene expression programs compared to their acute infection counterparts.

transcriptional program in responding CD4[+] T cells, which may be important to limit Th1-mediated immunopathology (*Penaloza-MacMaster et al., 2015*). This observation is in line with our previous finding that CXCR6[+] Th1, CXCR5[+] Tfh, and memory-like CD4[+] T cell subsets all display equivalent accessibility at key memory-associated gene loci, such as *Il7r*, *Tcf7*, *Slamf6*, and *Ccr7*, indicating that multiple distinct CD4[+] T cell lineages may remain poised to upregulate a memory-associated program during chronic infection (*Zander et al., 2022*). Additionally, and similar to CD8[+] T cells responding to chronic infection (*Wherry et al., 2007*), most CD4[+] T cell clusters from LCMV Cl13 infection uniformly displayed increased TCR signaling, T cell dysfunction, and T cell exhaustion gene expression profiles (*Figure 4F–G*, *Figure 4—figure supplement 1F*), once again suggesting that persistent exposure to antigen and inflammation may result in the upregulation of some conserved (and dysregulated) gene expression programs that span multiple phenotypically distinct CD4[+] T cell subsets. In line with this, all CD4[+] T cell clusters from LCMV Cl13 infection displayed notable increases in their expression of certain intracellular signaling phosphatases and kinases, such as *Dusp1*, *Dups2*, and *Pim1*, as well in their expression of several genes encoding various TFs, including *Jund*, *JunB*, *Fos*, *Klf2*, and *Bhlhe40* (*Figure 4I*). Additionally, genes known to be downstream of TCR signaling such as *Cd69*, *Nr4a2*, and *Tox* (*Alfei et al., 2019*; *Chen et al., 2019*; *Khan et al., 2019*), IFN-stimulated genes such as *Isg15* and *Irf7*, and genes encoding the inhibitory receptors Lag3, Tigit, and CTLA4 were all overexpressed across all CD4[+] T cell clusters from chronic infection compared to acute LCMV infection (*Figure 4I*). Together these data indicate that certain core transcriptional programs are upregulated across virus-specific CD4[+] T cell subsets independent of phenotype in response to sustained exposure to persistent viral infection. Conversely, some TFs with increased expression during chronic infection are more uniquely expressed by particular CD4[+] T cell subsets, such as *Zbtb32* in the Slamf6-hi cluster, *Eomes* in the Slamf7-hi cluster, and *Prdm1* (encodes Blimp1; *Figure 4J*) in the Th1 clusters from chronic infection, indicating that some gene expression programs are specific to a particular lineage. Of note, several genes or gene expression programs were also found to be downregulated in particular CD4[+] T cell subsets during chronic infection, such as *Id2*, *Cxcr6*, and effector CD8 T cell signature genes in Th1 cell clusters (*Figure 4J*, *Figure 4—figure supplement 1E*) as well as *Cxcr5* and *Icos* in the GC Tfh2 cluster, whereas Id3 appeared to be downregulated in most CD4[+] T cell subsets from chronic LCMV infection (*Figure 4J*). Collectively, these results demonstrate that chronic viral infection results in both altered CD4[+] T cell subset distribution as well as distinct gene expression programs, with CD4[+] T cell subsets displaying not only lineage-specific gene expression profiles but also core gene expression programs that are conserved across multiple distinct populations of T helper cells.

## Discussion

In this study, we demonstrate that CD4[+] T cells responding to chronic LCMV infection are more heterogeneous than previously appreciated, with several transcriptionally distinct subsets being detected within Th1, Tfh, and T memory-like CD4[+] T cell populations. We further demonstrate that while the progeny of most CD4[+] T cell clones displayed immense developmental plasticity, suggesting a one cell-multiple fate model, a small fraction of clones appeared to exhibit a biased cell fate-decision, indicating that TCR usage may have an impact on CD4[+] T cell differentiation. Additionally, we highlight some potential flow cytometric gating strategies that can be used to further interrogate these respective subsets and also provide evidence that the relative distribution of these populations differed substantially between acute and chronic viral infection. Lastly, our integrated analyses revealed that chronic viral infection results in vastly altered gene expression programs in responding CD4[+] T cells,

with evidence for not only lineage-specific gene expression profiles but also core gene expression programs that are upregulated and conserved across multiple distinct populations of T helper cells. Collectively, these data may serve as a useful resource to assess and compare the transcriptional landscape and differentiation trajectory of virus-specific CD4$^+$ T cell clones during acute and chronic viral infection.

Several recent studies have identified that CD4$^+$ T cells responding to acute viral infection are composed of multiple transcriptionally distinct CD4$^+$ T cell subsets, including Th1, Tfh, and Tcmp cells (*Ciucci et al., 2019*; *Khatun et al., 2021*). While the differentiation and helper functions of Th1 and Tfh subsets in these acute settings are fairly well established, recent work has also yielded important insight into the transcriptional profile and developmental biology of Tcmp cells. Notably, Tcmp cells display high expression of memory cell markers CCR7, TCF-1, and Bcl-2 and have an intrinsic capacity to survive the contraction phase of acute infection and generate diverse effector cell populations during a secondary response (*Ciucci et al., 2019*; *Marshall et al., 2011*; *Pepper and Jenkins, 2011*). The development of Tcmp cells has been demonstrated to be dependent on both Thpok and Bcl6 (*Ciucci et al., 2019*; *Pepper and Jenkins, 2011*). Thus, our understanding of CD4$^+$ T cell differentiation during acute infection is becoming increasingly more well defined. By comparison, our knowledge of how CD4$^+$ T cells adapt in the face of persistent antigen exposure and inflammation remains incompletely understood. Moreover, whether diverse populations of CD4$^+$ T cells also emerge during chronic infection or whether they can display memory-like properties have remained significant knowledge gaps in the field. Notably, recent studies have identified that a memory-like CD4$^+$ T cell population develops alongside Th1 and Tfh cell subsets during chronic viral infection (*Xia et al., 2022*; *Zander et al., 2022*) and that this memory-like subset bears a strikingly similar transcriptional profile as Tcmp cells from acute LCMV infection (*Zander et al., 2022*). However, whether additional heterogeneity exists amongst the CD4$^+$ T cell pool during chronic viral infection remains unclear. Moreover, an in-depth assessment into how the transcriptional landscape of CD4$^+$ T cells differs between acute and chronic viral infection at the single-cell level has not previously been explored. Herein, we demonstrate that CD4 T cells responding to chronic viral infection are more heterogeneous than previously appreciated, with several transcriptionally distinct subsets arising during the early phase of chronic infection. These include two potentially unique Th1 cell subsets (Ly6c-hi and Lag3-hi cells), two Tfh-like subsets (GC Tfh2 and Tfh1), a memory-like population, and a pre-Tfh subset that exhibits both a memory-like signature and also expresses some Tfh-associated genes. Additionally, our scRNA-seq analyses identified a cluster of Slamf7-hi cells that may represent a cytolytic CD4 T cell subset (*Cachot et al., 2021*; *Della-Torre et al., 2018*), as well as a unique cluster that displayed increased expression of multiple ISGs, consistent with chronic infection driving a sustained type I IFN response (*Teijaro et al., 2013*; *Wilson et al., 2013*). Using spectral flow cytometry, we validate the formation of several of these subsets during chronic viral infection and also highlight a few potential gating strategies and markers that may be useful in interrogating these subsets further. However, future work will be important to elucidate the developmental relationship between these respective populations and to establish whether some of these newly identified subsets represent functionally distinct lineages or merely transitional states along the Th1, Tfh, or memory T cell developmental pathways.

Notably, while our TCR clonal trajectory analyses indicate that the majority of GP66-specific CD4$^+$ T cell clones responding to chronic LCMV infection appear to be multipotent, a small fraction of T cell clones appeared to display a biased cell fate decision and favored a particular lineage. Overall, this finding is consistent with other recent studies from resolving infections (*Tubo et al., 2013*; *Khatun et al., 2021*) and suggests that the vast majority of naïve CD4$^+$ T cells are capable of giving rise to multiple distinct lineages independent of TCR sequence, although TCR structure may still influence the differentiation trajectory of some select clones. As certain CDR3 motifs from the TCR alpha chain have previously been implicated as being a useful predictor of T cell fate during acute LCMV infection (*Khatun et al., 2021*), a deeper investigation into whether particular TCR alpha or beta chain sequences impact CD4$^+$ T cell fate determination during chronic viral infection will be of future interest.

Consistent with chronic viral infection resulting in diminished Th1 differentiation and function (*Fuller et al., 2004*; *Brooks et al., 2005*; *Fahey et al., 2011*; *Yamada et al., 2016*), our integrated scRNA-seq analysis demonstrated that the proportions of both Ly6c-hi and Lag3-hi Th1 cells were dramatically reduced during chronic vs acute LCMV infection. Interestingly, module score analyses

identified that loss of the Th1 gene expression program was not restricted to only the Th1 cell subsets but was conserved across multiple distinct subsets, indicating that loss of Th1-like pro-inflammatory activity may be a generalizable feature of T helper cell responses during chronic viral infection. This loss in Th1 cell subset formation coincided with increased frequencies of multiple subsets, including Slamf6-hi, pre-Tfh, ISG, and Slamf7-hi clusters, whereas the Tfh clusters remained similar between infections. Intriguingly, we further observed an enriched quiescence or memory-associated transcriptional profile across most CD4⁺ T cell subsets responding to chronic viral infection, despite viral load remaining high at this time point. Although the reasons for this remain unclear, it is possible that CD4⁺ T cells responding to chronic infection need to acquire a more quiescent or memory-like phenotype in order to mitigate Th1-mediated pathology (*Penaloza-MacMaster et al., 2015*). This enhanced acquisition of a memory-like phenotype is supported by the relative increased formation of Slamf6⁺ and pre-Tfh populations during persistent infection, which exhibit high expression of *Tcf7*, *Il7r*, *Ccr7*, and *Bcl2*. However, it is important to note that despite this enriched memory-associated program, most CD4⁺ T cell subsets responding to chronic LCMV infection also displayed increased TCR signaling and T cell dysfunction or exhaustion module scores, indicating that these cell subsets are not likely to be in a truly quiescent state nor do they fit the description of canonical memory populations. Consistent with this, we identified that *Id3*, a TF known to be critical for memory T cell formation following acute infection (*Yang et al., 2011*), was substantially diminished in most CD4⁺ T cell subsets from chronic infection compared to their acute infection counterparts. Moreover, in line with increased TCR signaling during chronic viral infection, we observed a conserved increase in the expression of AP-1 family members *Jund* and *Junb*, as well as in *Cd69*, *Nr4a2*, and *Tox* gene expression across all CD4⁺ T cell subsets during LCMV Cl13 infection. Similarly, exhaustion markers *Lag3*, *Tigit*, and *Ctla4* were uniformly increased in all CD4⁺ T cell subsets responding to chronic viral infection. Intriguingly, while these data highlight a broad upregulation of dysfunctional programs spanning multiple distinct populations of CD4⁺ T cells during chronic viral infection, it is worthy to mention that Th1 cell subsets from chronic infection displayed markedly higher dysfunction and exhaustion module scores compared to all other subsets. This data, coupled with previous observations demonstrating a rapid loss of Th1 responses during chronic viral infection (which may possibly reflect clonal deletion) and that Th1 cells rapidly lose their capacity to produce pro-inflammatory cytokines as the infection progresses (*Fuller et al., 2004*; *Brooks et al., 2005*; *Fahey et al., 2011*; *Osokine et al., 2014*), indicate that Th1 cells do exhibit some cardinal features of the process of CD8⁺ T cell exhaustion. At the same time, however, Th1 cells (from both acute and chronic infection) displayed increased expression of effector CD8⁺ T cell genes (*Figure 4—figure supplement 1E*), and Th1 cells further exhibited the highest capacity to produce IFN-γ and TNF-α upon ex vivo peptide stimulation (*Zander et al., 2022*). Thus, while Th1 cells responding to chronic viral infection share some overlap in their transcriptional program with both effector and exhausted CD8 T cells, whether these cells (or other populations of CD4⁺ T cells) are truly 'exhausted' remains to be investigated further.

Collectively, our results highlight perturbed gene expression programs in CD4⁺ T cells during chronic viral infection, with apparent alterations in both lineage-specific T helper cell programs and also core expression modules that are differentially regulated across all CD4⁺ T cell subsets between acute and chronic viral infection. A future investigation into the transcriptional circuits underlying the altered CD4⁺ T cell differentiation that occurs during chronic infection may have important implications for strategies aimed at improving T cell-based immunotherapies during chronic infection.

## Materials and methods

**Key resources table**

| Reagent type (species) or resource | Designation | Source or reference | Identifiers | Additional information |
|---|---|---|---|---|
| Strain and strain background | Lymphocytic choriomeningitis virus (LCMV) Clone 13 (Cl13) strain | Rafi Ahmed Laboratory | Grew up in house | |
| Strain and strain background | LCMV Armstrong strain | Rafi Ahmed Laboratory | Grew up in house | |

*Continued on next page*

*Continued*

| Reagent type (species) or resource | Designation | Source or reference | Identifiers | Additional information |
|---|---|---|---|---|
| Antibody | APC/fire rat monoclonal anti-mouse CD4 (GK1.5) | Biolegend | Cat #100460 RRID:AB_2572111 | 1:300 |
| Antibody | AF647 rat monoclonal anti-mouse CD4 (GK1.5) | Biolegend | Cat #100460 RRID:AB_2572111 | 1:300 |
| Antibody | Pe/Cyanine7 rat monoclonal anti-mouse PD.1 (RMP1-30) | Biolegend | Cat #109110 RRID:AB_572017 | 1:200 |
| Antibody | APC/Cyanine7 rat monoclonal anti-mouse PD.1 (29F.1A12) | Biolegend | Cat #135224 RRID:AB_ 2563523 | 1:200 |
| Antibody | APC/Cyanine7 rat monoclonal anti-mouse/human CD44 (IM-7) | Biolegend | Cat #103028 RRID:AB_830785 | 1:1000 |
| Antibody | APC rat monoclonal anti-mouse CXCR5 (L138D7) | Biolegend | Cat #145506 RRID: AB_2561970 | 1:100 |
| Antibody | PE/Dazzle 594 rat monoclonal anti-mouse CXCR6 (SA051D1) | Biolegend | Cat #151117 RRID:AB_2721700 | 1:200 |
| Antibody | PerCPeFluor710 rat monoclonal anti-mouse CXCR6 (Danid2) | Thermofisher | Cat # 46-9186-82 RRID:AB_ AB_2734904 | 1:200 |
| Antibody | BV785 rat monoclonal anti-mouse Lag3 (C9B7W) | Biolegend | Cat # 125219 RRID:AB_256657 | 1:100 |
| Antibody | PE/Cyanine7 rat monoclonal anti-mouse CD127 (A7R34) | Biolegend | Cat #135014 RRID:AB_1937265 | 1:100 |
| Antibody | Brilliant Blue mouse monoclonal anti-mouse 700 Tim3 (5D12) | BD Biosciences | Cat #747619 RRID:AB_2744185 | 1:100 |
| Antibody | Spark NIR 685 Armenian hamster monoclonal anti-mouse CD69 (H1.2F3) | Biolegend | Cat #104557 RRID:AB_2860607 | 1:100 |
| Antibody | BUV395 rat monoclonal anti-mouse CD44 (IM7) | BD Biosciences | Cat #740215 RRID: AB_2739963 | 1:500 |
| Antibody | BV711 rat monoclonal anti-mouse PSGL1 (2PH1) | BD Biosciences | Cat #740746 RRID: AB_2740414 | 1:100 |
| Antibody | BV650 rat monoclonal anti-mouse CD83 (Michel-19) | BD Biosciences | Cat #740619 RRID: AB_2740317 | 1:100 |
| Antibody | BV605 rat monoclonal anti-mouse PD-1 (29 F.1A12) | Biolegend | Cat #135220 RRID: AB_2562616 | 1:200 |
| Antibody | BV510 mouse monoclonal anti-mouse Ly108 (13G3) | BD Biosciences | Cat #745073 RRID: AB_2742691 | 1:200 |
| Antibody | Percp cy5.5 mouse monoclonal anti-mouse CX3CR1 (SA011F11) | Biolegend | Cat #149010 RRID: AB_2564494 | 1:100 |
| Antibody | PerCP rat monoclonal anti-mouse Ly-6c (HK1.4) | Biolegend | Cat #128028 RRID: AB_10897805 | 1:100 |
| Antibody | Pacific Blue rat anti-mouse Bst2 | Biolegend | Cat #127108 RRID: AB_2028455 | 1:100 |
| Antibody | BV421 rat monoclonal anti-mouse Slamf7 (129C1) | BD Biosciences | Cat #747997 RRID: AB_2872458 | 1:100 |
| Antibody | Goat polyclonal Anti-mouse IL1R2 | Thermofisher | Cat # PA5-47759 RRID:AB_11152697 | 1:400 |
| Antibody | Pe/Cyanine5 syrian hamster monoclonal anti-mouse ICOS (15F9) | Biolegend | Cat # 107708 RRID: AB_313337 | 1:100 |
| Antibody | APC-Fire750 rat monoclonal anti-mouse OX40 (OX-86) | Biolegend | Cat #119423 RRID: AB_2715993 | 1:100 |

*Continued on next page*

*Continued*

| Reagent type (species) or resource | Designation | Source or reference | Identifiers | Additional information |
|---|---|---|---|---|
| Other | PE LCMV I-A(b) GP66-77 tetramer | NIH Tetramer Core Facility | https://tetramer.yerkes.emory.edu | 1:500 |
| Commercial assay or kit | Chromium Single Cell 3' Library & Gel Bead Kit v2 | 10× Genomics | Cat#PN-120267 | |
| Commercial assay or kit | Chromium Single Cell A Chip Kit | 10× Genomics | Cat#PN-1000009 | |
| Commercial assay or kit | Chromium i7 Multiplex Kit | 10× Genomics | Cat# PN-120262 | |
| Commercial assay or kit | Dynabeads MyOne Silane | Thermofisher | Cat#37002D | |
| Commercial assay or kit | SPRIselect Reagent Kit | Beckman Coulter | Cat#B23318 | |
| Commercial assay or kit | Kappa NGS quantification kit | KAPABiosystems | Cat#KK4824 | |
| Commercial assay or kit | NextSeq 500/550 High Output Kit v2.5 (150 cycles) | Illumina | Cat#20024907 | |
| Software and algorithm | Cell Ranger | 10× Genomics | https://support.10xgenomics.com/single-cell-gene-expression/software/pipelines/latest/installation | |
| Software and algorithm | Seurat | *Satija et al., 2015* | https://satijalab.org/seurat/ | |
| Software and algorithm | Monocle-2 | *Qiu et al., 2017*; *Trapnell et al., 2014* | http://cole-trapnell-lab.github.io/monocle-release/docs/ | |
| Software and algorithm | Flowjo Version 10.5.3 | Tree Star | N/A | |
| Software and algorithm | Prism 8 | Graphpad Software | N/A | |

## Mice and LCMV Cl13 infection

6–8-week-old female C57Bl/6 mice obtained through the National Cancer Institute grantees program (Frederick, MD) were used for all experiments, unless otherwise indicated. Mx1-GFP homozygous mice (Jackson strain #:033219) were crossed to C57Bl/6 mice to generate Mx1-GFP heterozygous mice that were then used for some experiments where indicated. Mice were bred and maintained in a closed breeding facility, and mouse handling conformed to the requirements of the Institutional Animal Care and Use Committee (IACUC) guidelines of the Medical College of Wisconsin. All of the animals were handled according to approved IACUC protocols (#'s 00003003 and 00003004) of the Medical College of Wisconsin. Mice were infected with $2\times10^6$ plaque forming units (PFU) of LCMV strain Cl13 via retroorbital injection to establish chronic viral infection.

## Flow cytometry and cell sorting

For cell sorting experiments, splenic CD4+ T cells from LCMV Cl13-infected mice were isolated using the EasySep mouse CD4 T cell isolation kit (STEMCELL; Cat#19852). Enriched CD4+ T cells were then stained using LCMV-specific $GP_{66-77}$ PE tetramer from National Institutes of Health along with CD4 and CD44 antibodies in FACS buffer. The staining was performed in the dark for 1 hr at room temperature, followed by three washes with FACS buffer. Gp66-specific CD4+ T cells were sort-purified on a FACS-Aria sorter. For FACS analyses, single cell suspensions obtained from spleenocytes from LCMV Cl13-infected mice were stained with $GP_{66-77}$ PE tetramer in conjunction with other surface antibodies in FACS buffer, followed by three washes in FACS buffer. Samples were then acquired on either an Aurora Cytek (Cytek Biosciences) or LSR-II (BD Biosciences). In some experiments, after surface staining, cells were fixed with buffer from the True-Nuclear TF Buffer Set (BioLegend) for 1 hr. Then cells were then washed with permeabilization buffer and stained with antibodies against TFs in permeabilization buffer. Analyses were performed using Flowjo software version 10.8.1.

## Single cell RNA sequencing

LCMV-specific ($GP_{66-77}$ tetramer+) CD4+ T cells were harvested from the spleens of two LCMV Cl13–infected mice on day 10 after infection and were FACS sorted using a FACS-Aria Sorter (BD Biosciences). Sorted cells were loaded onto the 10× Chromium Controller with a target cell number of 5000 per mouse. scRNA-seq libraries were prepared using the Chromium Single Cell 5' v2 Reagent Kit (10×

Genomics) according to the manufacturer's protocol. Two libraries were then quantified using the KAPA library quantification Kit and then were loaded onto an Illumina NextSeq 500 sequencer with the NextSeq 500/550 High Output Kit v2.5 (150 cycles; 20024907; Illumina) with the following conditions: 26 cycles for read 1, 98 cycles for read 2, and 8 cycles for the i7 index read. Raw sequencing data were downloaded from Illumina BaseSpace, then demultiplexed and converted to gene-barcode matrices using the 'mkfastq' and 'count' functions in Cell Ranger v3.0 (10× Genomics). A total of 4394 and 3613 cells were recovered from samples M1 and M2, respectively.

## Single cell TCR sequencing

A 10 µL aliquot of each sample's amplified cDNA was taken for scTCR-seq. The standard 10× protocol was followed for TCR gene amplification. Libraries were quantified using a KAPA library quantification kit (Roche Sequencing) and then loaded onto an Illumina NextSeq 500 sequencer using a NextSeq 500/550 High Output Kit v2.5 (150 cycle kit) (20024907, Illumina) with 150 cycles for read 1, 150 cycles for read 2, and 8 cycles for i7 index reads. Raw sequencing data were downloaded from Illumina BaseSpace, then demultiplexed and converted to gene-barcode matrices using the 'mkfastq' and 'vdj' functions in Cell Ranger v3.0 (10× Genomics).

## Combined analysis of scRNA-seq and scTCR-seq data

Analysis was primarily performed in R (v 3.6.1) using the package Seurat (v 3.1) (*Butler et al., 2018*; *Stuart et al., 2019*), with the package tidyverse (v 1.2.1) (*Wickham et al., 2019*) used to organize data and the package ggplot2 (v 3.2.1) used to generate figures. scRNA-seq datasets were integrated and then scTCR-seq data was added. scRNA-seq data were filtered to keep cells with a low percentage of mitochondrial genes in the transcriptome (<5%) and between 200 and 3000 unique genes to exclude poor quality reads and doublets. Cell cycle scores were regressed when scaling gene expression values and TCR genes were regressed during the clustering process, which was performed with the Louvain algorithm within Seurat and visualized with UMAP. Cells other than CD4$^+$ T cells, sample-specific outliers, and cells not belonging to clones with at least two cells were excluded. A clone was defined as a group of cells sharing the same TCR α and β chain CDR3 nucleotide sequences. Trajectory analyses were performed using Monocle (v 2.12.0; *Qiu et al., 2017*). The Monocle tree structure was generated using the DDRTree algorithm with the top 100 differentially expressed genes from each Seurat cluster.

## Gene set enrichment and module score analyses

A preranked analysis module in gene set enrichment analysis (GSEA) (*Subramanian et al., 2005*) was used, and the gene sets were found in MSigDB (*Liberzon et al., 2011*). For single cluster enrichment analysis, differentially expressed genes were identified first (logFC >0.1), and the average logFC expression of each gene was calculated. Then, this average gene expression data for each cluster was used as an input for GSEA. After this, a gene set variation analysis score (*Hänzelmann et al., 2013*) was calculated using log normalized expression data for each cluster and chosen gene sets, as identified by GSEA. Seurat clusters were scored by GSEA gene sets using the Addmodulescore function. Gene sets for Th1, GC T$_{FH}$, Tcmp, T cell dysfunction, and memory CD4 T cells were obtained from published data (*Ciucci et al., 2019*). Gene sets for progenitor, effector, and exhausted CD8 T cell subsets (*Zander et al., 2022*) or memory CD8 T cells (*Wherry et al., 2007*) were obtained from previously published data. Module scores for TCR signaling were calculated within our Seurat clusters using the 'TCR pathway' gene set from GSEA (KEGG hsa04660).

## Cellular distribution at the clonal level for three lineages

The cellular distribution of all clones (two or more cells/clone; n=327 clones) among the three different lineages defined by Monocle states (Th1, Tfh, and Slamf6 memory-like; described above) was shown using the pheatmap R package (*Figure 3E*). This allowed for visualization of cellular distributions of each clone among the lineages, thus demonstrating the differentiation preference of each clone (i.e. single fate or multifate). To statistically test whether some clones display a biased cell-fate decision, we performed an analysis using clones that have at least four cells (n=176 clones). Briefly, Th1 or Tfh biased clones were identified if 65% of the cells in that clone belonged to either the Th1 or Tfh lineage, respectively, based on their Monocle states (*Khatun et al., 2021*). The validation of this

biased clonal fate was performed using Th1 or Tfh module scores for each clone. Clonal Th1 or Tfh module score was calculated by taking the average module score for all cells in that clone. However, Th1 or Tfh module scores at the clonal level have a different scale or range so direct comparison at the clonal level was not possible. To do this, percentile ranks for the module scores at the clonal level were added by taking the average percentile module score for all cells in that clone (*Khatun et al., 2021*).

## Canonical correlation analysis

A comparative scRNA-seq transcriptomic analysis was performed by combining our previously published scRNA-seq data of $GP_{66}^+$ $CD4^+$ T cells isolated from two LCMV Armstrong-infected mice on day 10 p.i. (*Khatun et al., 2021* GSE 158896) with our current dataset of $GP_{66}^+$ CD4 T cells from day 10 post-LCMV Cl13 infection. To do this, we used the integration function in Seurat to perform a CCA (*Butler et al., 2018*) and identify shared subpopulations across datasets. As an initial step of the analysis, cells were filtered based on having high percentage of mitochondrial genes in the transcriptome (>10%) and less than 200 to ~4000–5000 unique genes to remove any doublets and dead cells. Following this, a total of ~18,000 cells were included in the analysis for four mice. Cell cycle gene scoring was calculated for all cells and regressed out. To understand cellular heterogeneity, unsupervised clustering was performed using a dimension reduction algorithm called, UMAP, based on 3000 highly variable genes with an input of the top 20 principle components.

## Statistical analysis

Statistical tests for flow cytometry data were performed using Graphpad Prism 8. p Values were calculated using two-tailed unpaired Student's t-tests. p Values for violin plots showing gene set-specific module scores across clusters were calculated by the Wilcoxon test with Holm-Sidak correction.

## Acknowledgements

This work is supported by NIH grants AI125741 (WC), AI148403 (WC), DK127526 (MYK), AI153537 (RZ), and by an American Cancer Society (ACS) Research Scholar Grant (WC); and by an Advancing a Healthier Wisconsin Endowment (AHW) Grant (WC). RZ also received support from a Cancer Research Institute Irvington Fellowship. MYK is a member of the Medical Scientist Training Program at the Medical College of Wisconsin (MCW), which is partially supported by a training grant from NIGMS (T32-GM080202). This research was completed in part with computational resources and technical support provided by the Research Computing Center at MCW.

## Additional information

### Funding

| Funder | Grant reference number | Author |
|---|---|---|
| NIH NIAID | R01 AI125741 | Weiguo Cui |
| NIH NIAID | K99/R00 AI153537 | Ryan Zander |
| NIH NIAID | R01 AI148403 | Weiguo Cui |
| NIH NIAID | DK127526 | Moujtaba Y Kasmani |
| American Cancer Society | | Weiguo Cui |
| NIGMS | T32-GM080202 | Moujtaba Y Kasmani |

The funders had no role in study design, data collection and interpretation, or the decision to submit the work for publication.

### Author contributions

Ryan Zander, Conceptualization, Data curation, Software, Formal analysis, Validation, Investigation, Visualization, Writing - original draft, Writing - review and editing; Achia Khatun, Conceptualization, Data curation, Software, Formal analysis, Visualization, Methodology, Writing - review and editing;

Moujtaba Y Kasmani, Software, Formal analysis, Writing - review and editing; Yao Chen, Data curation, Investigation; Weiguo Cui, Conceptualization, Resources, Supervision, Funding acquisition, Writing - review and editing

**Author ORCIDs**
Ryan Zander ![ORCID] http://orcid.org/0000-0002-7431-8535
Moujtaba Y Kasmani ![ORCID] http://orcid.org/0000-0002-5753-5335
Weiguo Cui ![ORCID] http://orcid.org/0000-0003-1562-9218

**Ethics**
This study was performed in strict accordance with the recommendations in the Guide for the Care and Use of Laboratory Animals of the National Institutes of Health. Mice were bred and maintained in a closed breeding facility, and mouse handling conformed to the requirements of the Institutional Animal Care and Use Committee guidelines of the Medical College of Wisconsin. All of the animals were handled according to approved institutional animal care and use committee (IACUC) protocols (#'s 00003003 & 00003004) of the Medical College of Wisconsin.

**Decision letter and Author response**
Decision letter https://doi.org/10.7554/eLife.80079.sa1
Author response https://doi.org/10.7554/eLife.80079.sa2

## Additional files

**Supplementary files**
• MDAR checklist

• Supplementary file 1. Clonal Distribution and cellular fate of GP66-specific CD4 T cell clones (with 4 or more cells per clone) during LCMV Cl13 infection.

**Data availability**
The scRNA-seq and scTCR-seq data have been deposited in the GEO database (accession no GSE201730), and are available to the public.

The following dataset was generated:

| Author(s) | Year | Dataset title | Dataset URL | Database and Identifier |
|-----------|------|---------------|-------------|--------------------------|
| Cui W, Zander R, Khatun A | 2022 | Single-cell RNA-sequencing combined with TCR-sequencing of GP66+ splenic CD4+ T cells on day 10 post LCMV Cl13 infection | https://0-www-ncbi-nlm-nih-gov.brum.beds.ac.uk/geo/query/acc.cgi?acc=GSE201730 | NCBI Gene Expression Omnibus, GSE201730 |

The following previously published dataset was used:

| Author(s) | Year | Dataset title | Dataset URL | Database and Identifier |
|-----------|------|---------------|-------------|--------------------------|
| Cui W, Khatun A | 2020 | Single-cell lineage mapping of a diverse virus-specific naïve CD4 T cell repertoire | https://www.ncbi.nlm.nih.gov/geo/query/acc.cgi?acc=GSE158896 | NCBI Gene Expression Omnibus, GSE158896 |

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
