## [Editor Report]

The present study by Zander et al., aims at improving our understanding of CD4^+^ T cell heterogeneity in response to chronic viral infections. The authors utilize the murine LCMV c13 infection model and perform single-cell RNA seq analysis on day 10 post-infection to identify multiple, previously unappreciated, T-cell subsets and then go on and verify these analyses using multi-color flow cytometry before comparing the transcriptome of CD4 T cells from chronic infection to a previously generated data set of CD4 T cells obtained from acutely-resolved LCMV infection. This is a valuable transcriptomic resource of murine CD4 T cell subsets in chronic viral infection. The study will be of broad interest to a wide range of researchers focused on studying CD4 T cell biology.

---

## [Decision Letter]

**Decision letter after peer review:**

Thank you for submitting your article "Delineating the transcriptional landscape and clonal diversity of virus-specific CD4 + T cells during chronic viral infection" for consideration by *eLife*. Your article has been reviewed by 2 peer reviewers, and the evaluation has been overseen by a Reviewing Editor and Carla Rothlin as the Senior Editor. The reviewers have opted to remain anonymous.

Essential revisions:

1) Increase the statistical rigor of the analyses by correcting for biases as described by the reviewers.

2) Study and describe the biological basis for the tree branching identified via monocle

3) Show data to support that LCMV is cleared by day 10, or provide additional data that extends past day 10.

*Reviewer #1 (Recommendations for the authors):*

Congratulations on a very nice study.

The study would clearly benefit from an extension of the comparison of CD4 T cell subsets from chronic to acute infection. The authors utilize a 'dysfunction' geneset from 2007, which clearly is outdated. The authors could even utilize their own datasets from of DEGs from effector cells from chronic vs acute infection. Additionally, the authors could look at the enrichment score of genes highly expressed in TCF1+ precursor CD8^+^ T cells compared to equivalent cells from acute infection.

Also, the illustration of these module scores is not too intuitive. An alternative could make it easier for the reader to immediately recognize some differences, especially since it is such an interesting analysis. Are some subsets 'exhausted' while others are not?

Moreover, the authors could even further expand this section and identify signatures that distinguish Th1, Tfh or Tcmp subsets in both infections.

Finally, it would be nice if the authors could further integrate their findings into the current concept of CD4 T cell exhaustion.

*Reviewer #2 (Recommendations for the authors):*

– The paper should be better contextualized in the introduction with regards to the prior publications from the same group (Khatun 2021, Zander 2022). This will help the reader focus on what is new within the current work.

– Is there any biological basis for a tree structure for the lineage relationships of the different cell subtypes such as assumed by the monocle analysis?

– Based on the displayed plots in Figure 3AB Monocle seems to have identified three further branching points (labelled 1-3 in the plot) beyond the main branching point (labelled 4 in the plot). This is in contradiction with the description of the tree in the text as being tripartite and deserves some discussion.

– To demonstrate biases in clonal fate choice beyond chance a more robust statistical analysis is needed. In a clone of two cells say even by chance both of these might come from the same phenotypic cluster even in the absence of bias. Therefore the stated proportions of clones with broad or more narrow fates are not a meaningful metric.

– In the clonal trajectory analysis biases in clonal fates are not necessarily indicative of TCR-signaling driven fate choice, as TCR clones are also linked by a common lineage relationship. Stronger evidence for TCR signaling might be provided if it can be demonstrated that fate choices are correlated across distinct clones that have TCRs with a similar sequence, an approach the authors have taken in prior work but is missing here.

---

## [Author Response]

Essential revisions:1) Increase the statistical rigor of the analyses by correcting for biases as described by the reviewers.2) Study and describe the biological basis for the tree branching identified via monocle3) Show data to support that LCMV is cleared by day 10, or provide additional data that extends past day 10.

We sincerely thank the editors and reviewers for their useful feedback and careful consideration of our manuscript. We believe that we have sufficiently addressed the major concerns of the reviewers, and that our study has been substantially improved following these revisions. Please find our point-by-point response to each individual comment below.

Reviewer #1 (Recommendations for the authors):Congratulations on a very nice study.The study would clearly benefit from an extension of the comparison of CD4 T cell subsets from chronic to acute infection. The authors utilize a 'dysfunction' geneset from 2007, which clearly is outdated. The authors could even utilize their own datasets from of DEGs from effector cells from chronic vs acute infection. Additionally, the authors could look at the enrichment score of genes highly expressed in TCF1+ precursor CD8^+^ T cells compared to equivalent cells from acute infection.Also, the illustration of these module scores is not too intuitive. An alternative could make it easier for the reader to immediately recognize some differences, especially since it is such an interesting analysis. Are some subsets 'exhausted' while others are not?Moreover, the authors could even further expand this section and identify signatures that distinguish Th1, Tfh or Tcmp subsets in both infections.Finally, it would be nice if the authors could further integrate their findings into the current concept of CD4 T cell exhaustion.

We thank the reviewer for these helpful suggestions. We have now performed additional module score analyses, of which include a comparison of progenitor, effector and exhausted CD8 T cell gene expression programs (Zander et al., 2019) across the CD4 T cell subsets identified from our canonical correlation analysis of virus-specific CD4 T cells responding to acute and chronic LCMV infection. In addition to displaying the relative “score” of these expression programs, we have now additionally included a dot plot (in Figure 4—figure supplement 1F) depicting key signature genes from these expression modules, so that readers can assess the relative expression of indicated genes across CD4 T cell clusters.

With regards to the reviewer’s question on whether certain subsets appear more “exhausted” than others, we do in fact observe that Th1 cell subsets from chronic LCMV infection display markedly higher dysfunction and exhaustion module scores compared to their Th1 counterparts from acute LCMV infection, as well as all other CD4 T cell clusters (Figure 4G-H). Additionally, most CD4 T cell clusters responding to chronic LCMV infection displayed relatively increased dysfunction scores compared to their acute infection counterparts. In line with these findings, we observed that the relative expression of several exhausted and dysfunctional-associated genes (*Pdcd1*, *Lag3, Tox, Nr4a2, Tigit* and *Ctla4*) were upregulated in most CD4 T cell clusters from chronic LCMV infection compared to their acute infection counterparts, and many of these genes were highly expressed in the Th1 cell subsets from chronic infection (Figures 4I, and Figure 4—figure supplement 1F). Taken together, these results show that Th1 cells from chronic infection appear to exhibit a similar gene expression program as exhausted CD8 T cells, although whether these cells are truly “exhausted” remains to be investigated further. Of note, however, and consistent with an “exhausted” phenotype, previous work has demonstrated that there is a rapid loss of Th1 responses during chronic viral infection (which may possibly reflect clonal deletion), and that these cells further display a diminished capacity to produce pro-inflammatory cytokines as the infection progresses. However, Th1 cells still appear to be the major producers of IFN-g and TNF-a during chronic infection, suggesting that these cells may still might play an important role in contributing to control over viral replication.

As suggested by the reviewer, we have included further commentary on these findings in the Discussion section in order to further integrate these findings into the current concept of CD4 T cell exhaustion.

Reviewer #2 (Recommendations for the authors):– The paper should be better contextualized in the introduction with regards to the prior publications from the same group (Khatun 2021, Zander 2022). This will help the reader focus on what is new within the current work.

We have now expanded some of the paragraphs in the introduction section to help contextualize previous findings from our group and others, and also state what some of the current knowledge gaps in the field were that this study sought to address. These sections in the revised manuscript have been highlighted in yellow.

– Is there any biological basis for a tree structure for the lineage relationships of the different cell subtypes such as assumed by the monocle analysis?

This is a useful point brought up by the reviewer, and yes, we do believe that there is a biological basis for this tree structure. We and others have recently demonstrated that a subset of Slamf6+ TCF-1^hi^ memory-like CD4 T cells display progenitor-like qualities and can differentiate into either Th1 or Tfh cell subsets during persistent viral infection (Zander et al., *Immunity* 2022, Xia et al., *Immunity* 2022). Similarly, several groups have previously demonstrated that Slamf6 (Ly108) expression marks a subset of progenitor CD8 T cells that give rise to more terminally differentiated subsets during persistent infection and cancer. (Miller et al., *Nature Immunology* 2019, Utzschneider et al., *Immunity* 2016, Zander et al., *Immunity* 2019). For this reason, the Slamf6+ memory-like subset was chosen to serve as the root state for our Monocle analyses. Interestingly, memory-like cells appeared to fall along all major branch points, whereas Th1 cell subsets and Tfh cell subsets appeared to localize on separate branches, consistent with these later populations representing separate T cell lineages (Figure 3A). This observation is also consistent with our previous adoptive transfer experiments showing that Th1 and Tfh cells do not readily interconvert between one another following transfer into infection-matched hosts (Zander et al., 2022). We have now discussed these observations further under the trajectory analysis section of the revised manuscript.

– Based on the displayed plots in Figure 3AB Monocle seems to have identified three further branching points (labelled 1-3 in the plot) beyond the main branching point (labelled 4 in the plot). This is in contradiction with the description of the tree in the text as being tripartite and deserves some discussion.

The reviewer raises an interesting point. In our initial Monocle analysis, which used the top 110 significant differentially expressed genes to construct the tree, we described the tree as tripartite since there were three major branches present that contained either Th1 cells, Tfh cells, or memory-like cells. However, as mentioned by the reviewer, three smaller branches were also present, and these were located along the branch way spanning from the memory-like cells towards the Tfh cell clusters. Some pre-Tfh cells, Slamf7-hi cells, Tfh cells, and memory-like cells could be seen residing on these smaller branches as well as on the main branch along this trajectory. Overall, we believe that these observations are consistent with there being multiple transcriptionally distinct CD4 T cell populations being present during chronic viral infection, and this may further reflect how far a particular cluster has progressed along a developmental pathway, with cells at the end of branches appearing to be more terminally differentiated than the cells located along the middle of the major branches. For example, many memory-like cells and pre-Tfh cells can be seen along the middle of the two major branches, but not as much at the tips of the branches, suggesting that these populations may exhibit more developmental plasticity compared to the other cells, such as Th1 cells that reside at the end of the branch and accordingly have likely progressed to a terminal differentiation state.

Of note, we have since re-ran our Monocle analysis using only the top 100 significant DEGs to construct the tree, which has resulted in a more simplified tree structure that once again still has three major branches, but only contains one smaller branchpoint that appears to be enriched with pre-Tfh cells, but also contains some Tfh cells and memory-like cells. Overall, the findings remain very similar to our initial analysis in that mature Th1 and Tfh cells are found at the end of the branches, whereas the Pre-Tfh and memory-like cells are more centrally located along multiple branches. We have now discussed these observations more thoroughly in the paper, and the changes have been highlighted in yellow.

– To demonstrate biases in clonal fate choice beyond chance a more robust statistical analysis is needed. In a clone of two cells say even by chance both of these might come from the same phenotypic cluster even in the absence of bias. Therefore the stated proportions of clones with broad or more narrow fates are not a meaningful metric.

The reviewer raises an excellent point. In our previous analysis, we were simply using the ≥ 2 cells per clone cutoff to provide an overall view of the cellular distribution for different cell fates. We agree that in order to make any conclusions on the presence of biased cell fate decisions, a more stringent cut-off (higher number of cells/clone) would be required.

Thus, to test whether some clones display a biased cell fate decision, we have now performed an analysis using clones that have at least 4 cells, which left 176 CD4 T cell clones for analysis. The clonal distribution and cellular fate of these 176 clones are plotted in the heatmap (Figure 3—figure supplement 1A; similar to Figure 3E). To determine clones with biased cellular fates, a threshold of 65% was used meaning Th1 or TFH biased clones were identified if 65% of the cells in that clone belonged to either the Th1 or TFH lineage, respectively, based on their Monocle states (Khatun et al., 2021). The validation of this biased clonal fate was performed using Th1 or TFH module scores for each clone. Clonal Th1 or TFH module score was calculated by taking the average module score for all cells in that clone. However, Th1 or TFH module scores at the clonal level have a different scale or range so direct comparison at the clonal level was not possible. To do this, percentile ranks for the module scores at the clonal level was added by taking the average percentile module score for all cells in that clone (Khatun et al., JEM 2021).

Notably, and similar to our analysis using at least 2 cells per clone, we identified that a major proportion of clones appear multi-fated. Importantly, however, a small proportion of clones appeared to display a biased cell fate decision, with approximately 21% of the 176 clones displaying a preferential skewing towards the Th1 lineage (p = 2.443521e-09), and approximately 8.5% showing a preferential bias for a Tfh cell fate (p=3.903672e-05). These results are now displayed in Figure 3—figure supplement 1, and additions to the text are highlighted in yellow.

– In the clonal trajectory analysis biases in clonal fates are not necessarily indicative of TCR-signaling driven fate choice, as TCR clones are also linked by a common lineage relationship. Stronger evidence for TCR signaling might be provided if it can be demonstrated that fate choices are correlated across distinct clones that have TCRs with a similar sequence, an approach the authors have taken in prior work but is missing here.

We agree with the reviewer’s point, and we wanted to clarify that we are not trying to make any conclusions about the impact of TCR-signaling on regulating cell fate decisions here, but rather wanted to show that TCR signaling appears augmented in multiple developmentally distinct CD4 T cell subsets during chronic viral infection compared to acute infection. This is evident through the augmented TCR signaling module scores spanning multiple subsets, and also the upregulation of multiple genes downstream of TCR signaling indicative of sustained TCR stimulation in the presence of chronic viral infection. However, a determination of how persistent TCR signaling shapes CD4 T cell development and function during chronic viral infection remains an ongoing investigation in our lab.